



Earth System
Dynamics

# Diurnal land surface energy balance partitioning estimated from the thermodynamic limit of a cold heat engine

**Axel Kleidon and Maik Renner**

Biospheric Theory and Modelling Group, Max-Planck-Institut für Biogeochemie, Jena, Germany

**Correspondence:** Axel Kleidon (akleidon@bgc-jena.mpg.de)

**Abstract.** Turbulent fluxes strongly shape the conditions at the land surface, yet they are typically formulated in terms of semiempirical parameterizations that make it difficult to derive theoretical estimates of how global change impacts land surface functioning. Here, we describe these turbulent fluxes as the result of a thermodynamic process that generates work to sustain convective motion and thus maintains the turbulent exchange between the land surface and the atmosphere. We first derive a limit from the second law of thermodynamics that is equivalent to the Carnot limit but which explicitly accounts for diurnal heat storage changes in the lower atmosphere. We call this the limit of a "cold" heat engine and use it together with the surface energy balance to infer the maximum power that can be derived from the turbulent fluxes for a given solar radiative forcing. The surface energy balance partitioning estimated from this thermodynamic limit requires no empirical parameters and compares very well with the observed partitioning of absorbed solar radiation into radiative and turbulent heat fluxes across a range of climates, with correlation coefficients $r^2 \geq 95\%$ and slopes near 1. These results suggest that turbulent heat fluxes on land operate near their thermodynamic limit on how much convection can be generated from the local radiative forcing. It implies that this type of approach can be used to derive general estimates of global change that are solely based on physical principles.

## 1 Introduction

The turbulent fluxes of sensible and latent heat play a critical role in the land surface energy balance during the day as these fluxes represent the principal means by which the surface cools and exchanges moisture, carbon dioxide and other compounds with the atmosphere. Due to their inherently complex nature, these fluxes are typically described by semiempirical expressions (e.g., Businger et al., 1971; Louis, 1979; Beljaars and Holtslag, 1991). Yet representations of this exchange in land surface and climate models are still associated with a high degree of uncertainty. This uncertainty results, for instance, in biases in evapotranspiration and surface temperatures across different models (Mueller and Seneviratne, 2014), in empirical relationships of land surface exchange outperforming land surface models (Best et al., 2015), and in biases in boundary layer heights (Davy

and Esau, 2016). The semiempirical and highly coupled nature of land–atmosphere exchange seems to make it almost impossible to derive simple, physically based estimates of the magnitude of turbulent exchange and how it changes with land cover change or global warming.

An alternative approach to describing surface–atmosphere exchange can be based on thermodynamics (Kleidon et al., 2014; Dhara et al., 2016), an aspect that is rarely considered in the description of surface–atmosphere exchange. In this approach, turbulent exchange is formulated as a thermodynamic process by which turbulent heat fluxes drive a convective heat engine within the atmosphere that does the work to maintain convection and thus the turbulent exchange near the surface. This approach specifically invokes the second law of thermodynamics as an additional constraint on atmospheric dynamics (similar to previous approaches, such as the maximization of material entropy production (MEP); e.g., Pal-

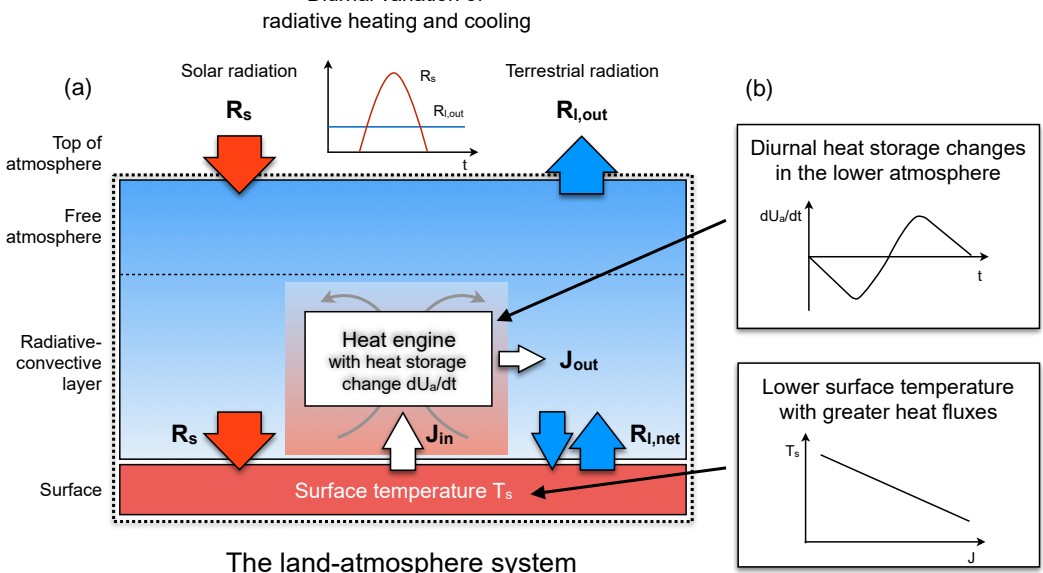

**Figure 1.** Schematic diagram of the land–atmosphere system where turbulent heat fluxes from the surface, $J_{in}$, act as the driver of an atmospheric heat engine that generates convective motion and which sustains the heat fluxes. The heat source of the engine is the absorption of solar radiation at the surface, $R_s$, reduced by the net exchange of terrestrial radiation, $R_{l,net}$, which depends on surface temperature. The two critical effects that set the limit on how much work the engine can perform are illustrated in panel (**b**): diurnal changes in heat storage in the lower atmosphere due to the diurnal variation of solar radiation and the reduction in surface temperature, $T_s$, due to greater turbulent heat fluxes both lower the work output of the engine.

tridge, 1978; Ozawa and Ohmura, 1997; Lorenz et al., 2001; Ozawa et al., 2003). The second law sets a limit on how much work can be derived from the local radiative forcing of the system. The dynamics associated with convection are then essentially captured by the implicit assumption that convection works as hard as it can, so that the use of the thermodynamic limit approximates the emergent convective dynamics. Previous applications of this thermodynamic approach have shown that it can successfully describe the broad climatological variation of surface energy balance partitioning on land and ocean (Kleidon et al., 2014; Dhara et al., 2016), the strength and sensitivity of the hydrologic cycle and surface temperatures to global change (Kleidon and Renner, 2013a, b, 2017; Kleidon et al., 2015), and the dynamics of the Earth system in general (Kleidon, 2016).

Here we extend this approach to the diurnal variation of the surface energy balance on land and compare its estimated partitioning to observations across different climates. As in the previous applications of thermodynamics to land–atmosphere exchange, the starting point is to view turbulent fluxes as the result of a heat engine that is driven by these heat fluxes (Fig. 1). The limit on how much work this heat engine can maximally perform is set by the first and second law of thermodynamics, from which the well-known Carnot limit of a heat engine can be derived (e.g., Kleidon, 2016).

When applied to the setting of the diurnal cycle of the land–atmosphere system, two key aspects need to be consid-

ered as these shape the thermodynamic limit (as illustrated by the two boxes in Fig. 1b). First, the strong diurnal variation of solar radiation causes strong changes in heat storage within the system that result in a much less varying emission of terrestrial radiation to space. In the absence of such heat storage changes, nighttime temperatures would be much lower than those found on Earth. In the ideal case that is being considered here, the strong variation of solar radiation is completely leveled out to yield a uniform emission of radiation to space, as indicated by the blue line in the graph at the top of Fig. 1 labeled $R_{l,out}$. While these heat storage changes predominantly take place below the surface for open water surfaces such as the ocean and lake systems (reflected in nearly uniform turbulent fluxes during night and day; see, e.g., measurements by Liu et al., 2009), the land–atmosphere system accommodates these changes mostly in the lower atmosphere (Kleidon and Renner, 2017) because heat diffusion into the soil is slow (e.g., Oke, 1987). The relevance of this different way of accommodating heat storage changes over land is that it takes place within the heat engine that we consider. The heat storage change is associated with a heating of the engine during the day, which represents an additional term in the entropy balance of the engine. What we show here is that the resulting thermodynamic limit is somewhat different to the common Carnot limit. We refer to this limit as the Carnot limit of a cold heat engine. Our motivation to refer to this limit as the limit of a cold heat engine is the be-

havior of a cold car engine in winter. When the car engine is still cold just after it has been started, one needs to hit the gas pedal harder to get the same power. As we will show below, the expression we derive here shows the same effect, that is, that a heat gain inside the engine reduces the work output of the engine. We will show that this enhanced heat flux is consistent with observations, so that this effect of heat accumulation during the day is an important factor that shapes the magnitude of turbulent fluxes on land.

The magnitude of the diurnal variation in heat storage is well constrained when assuming that the radiative heating by solar radiation and the emission to space are roughly balanced over the course of day and night. The temporal change in heat storage during the day can then be inferred from the imbalance of radiative fluxes at the top of the atmosphere (indicated in the upper panel of Fig. 1b, and as described by Kleidon and Renner, 2017).

The second aspect that shapes the thermodynamic limit is the reduction in surface temperature in the presence of greater turbulent fluxes at the surface (lower panel at the right of Fig. 1). This reduction in surface temperature reduces the temperature difference that is utilized by the heat engine to derive power, thus setting a limit of maximum power of the heat engine (as in, e.g., Kleidon and Renner, 2013a; Kleidon et al., 2014; Dhara et al., 2016). (This maximum power limit is very closely related to the proposed principle of maximum entropy production (MEP), as maximum power equals maximum dissipation in steady state, and entropy production is proportional to dissipation. An example of the application of MEP to convection is given by Ozawa and Ohmura, 1997.) We then combine the thermodynamic limit of a cold heat engine with the energy balances of the surface and of the whole surface–atmosphere system and maximize the power output of the heat engine to get a fully constrained description of the system that can, in first approximation, be solved analytically. It yields a description of the turbulent exchange between the land surface and the atmosphere that is fully constrained by thermodynamics and free of empirical turbulence parameterizations.

In the following, we first derive the thermodynamic limit of a cold heat engine, combine it with the energy balances of the system and maximize the power output to estimate surface energy balance partitioning based on the solar forcing of the system. The estimated partitioning is then tested with observations across field sites of contrasting climatological conditions. We then discuss how our thermodynamic approach compares to the common approaches in boundary layer meteorology and consider the utility of our approach for future work as well as potential implications.

## 2 Thermodynamic formulation of the land surface energy balance

We consider the land–atmosphere system as a thermodynamic system in a steady state when averaged over the diurnal cycle. Surface heating by absorption of solar radiation, $R_s$, causes the surface to warm, while the atmosphere is cooled by the emission of radiation to space, $R_{l,out}$ (Fig. 1). The surface and atmosphere are linked by the net exchange of terrestrial radiation, $R_{l,net}$, and turbulent heat fluxes, $J_{in}$, that result from convective motion. We consider this system to be a locally forced system with no advection. Convective motion within the boundary layer is seen as the consequence of a heat engine that generates motion out of the turbulent heat fluxes, where, for simplicity, we do not distinguish between the effects of the sensible and latent heat flux and the associated forms of dry and moist convection. The steady-state condition is used for the radiative forcing of the whole system by requiring that the mean radiative fluxes taken over the whole day are balanced such that $R_{s,avg} = R_{l,out}$ (with $R_{s,avg}$ being the average of $R_s$). Furthermore, we assume that the generation of turbulent kinetic energy, or power $G$ (or work per time), and its frictional dissipation, $D$, are in balance, so that $G = D$. In the following, we derive the limit on how much power can be derived from the forcing of the system directly from the first and second law of thermodynamics in a general way, so that we do not need to make the assumption that the atmosphere operates in a Carnot-like cycle. All variables used in the following are summarized and described in Table 1.

### 2.1 Carnot limit with heat storage changes

We first derive a thermodynamic limit akin to the Carnot limit from the energy and entropy balances of the heat engine, which specifically includes the change in heat storage within the engine. The first law of thermodynamics applied to this setup is given by

$$\frac{dU_e}{dt} = J_{in} + D - J_{out} - G, \qquad (1)$$

where $dU_e/dt$ is the change in heat storage within the heat engine, $J_{in}$ represents the addition of heat by the turbulent heat fluxes from the surface and $J_{out}$ is the rate by which the heat engine is being cooled, which is accomplished by radiative cooling. Note that this formulation differs from the derivation of the Carnot limit by accounting for changes in internal energy on the left-hand side and for dissipative heating, $D$, on the right-hand side as frictional dissipation takes place within the system. As we consider a steady state with $G = D$, note that the contributions of these terms in Eq. (1) cancel out so that the equation reduces to $dU_e/dt = J_{in} - J_{out}$. Also note that at this point, we neglect the effects of radiative energy transport from the surface to the atmosphere that would contribute to $dU_e/dt$ in the application to the surface–

atmosphere system. As it turns out, this contribution by radiation does not alter the limit, as shown in Appendix A.

The associated entropy budget of the heat engine is given by a change in entropy associated with the change in heat storage, $dU_e/dt$, at an effective engine temperature $T_e$, the entropy input by $J_{in}$ at a temperature $T_s$, the entropy export by $J_{out}$ at a temperature $T_a$, frictional dissipation that is assumed to occur at temperature $T_e$, and possibly some irreversible entropy production $\sigma_{irr}$ within the engine, i.e. irreversible losses that are not accounted for by the frictional dissipation term, $D/T_e$. TS1 CE1

$$\frac{1}{T_e}\frac{dU_e}{dt} = \frac{J_{in}}{T_s} + \frac{D}{T_e} - \frac{J_{out}}{T_a} + \sigma_{irr}. \qquad (2)$$

Note that this entropy budget is the entropy budget for thermal entropy, not for radiative entropy. This is an important distinction. A contribution by a radiative flux, e.g., a flux $R_{l,out}/T_a$, represents a flux of radiative entropy (and would require an additional factor of 4/3 as it deals with radiation); i.e., it is entropy reflected in the composition of radiation but not associated with the thermal motion of molecules that describes heat or thermal energy. As we deal with a convective heat engine, we must not include radiative terms as such but only when radiation is absorbed and heats air and water (adds thermal energy) or when the net emission of radiation cools (removes thermal energy). Radiative terms and radiative entropy production are typically much larger in the Earth system than non-radiative contributions (easily by a factor of 100, e.g., Kleidon, 2016). Yet any form of motion is associated with the much smaller but relevant thermal entropy terms.

For the atmospheric temperature, $T_a$, we use the radiative temperature associated with $R_{l,out}$ (i.e., we use the Stefan–Boltzmann law, $R_{l,out} = \sigma T_a^4$, to infer $T_a$, with $\sigma = 5.67 \times 10^{-8}\,W\,m^{-2}\,K^{-4}$ being the Stefan–Boltzmann constant). This is the most optimistic temperature for the entropy export from the heat engine as it is the coldest temperature possible to emit radiation at a rate $R_{l,out}$ to space, and it thus represents the highest entropy export from the heat engine (note that blackbody radiation represents the radiative flux with maximum entropy). Note also that this temperature is not bound to a particular height within the atmosphere but is instead inferred from the energy balance constraint. The effective engine temperature, $T_e$, essentially represents the potential temperature of the lower atmosphere as the temperature variation within the lower atmosphere is shaped by convection and is thus approximately adiabatic.

The thermodynamic limit on how much power, $G$, can maximally be derived by the engine is obtained from the entropy budget using the ideal case in which $\sigma_{irr} = 0$ (the second law of thermodynamics requires $\sigma_{irr} \geq 0$). This ideal case implies that the only source of entropy production is the frictional dissipation term, $D/T_e$ (cf. Eq. 2). TS2 CE2 Using

Eq. (1) to replace $J_{out}$ in Eq. (2), we obtain

$$G = J_{in} \cdot \frac{T_e}{T_s} \cdot \frac{T_s - T_a}{T_a} - \frac{dU_e}{dt} \cdot \frac{T_e - T_a}{T_a}. \qquad (3)$$

In this expression, the temperature of the heat engine, $T_e$, plays an important role. In the limiting case of $T_e \approx T_a$, this expression reduces to the common Carnot limit as the effect of the change in heat content is indistinguishable from the waste heat flux, $J_{out}$, of the heat engine. As the engine temperature essentially represents the potential temperature of the lower atmosphere, it is much closer to the surface temperature, so that the approximation $T_e \approx T_s$ is better justified. With this approximation, the thermodynamic limit of power then reduces to

$$G \approx \left(J_{in} - \frac{dU_e}{dt}\right) \cdot \frac{T_s - T_a}{T_a}. \qquad (4)$$

In the absence of heat storage changes, the term $dU_e/dt$ vanishes and yields, again, the common Carnot limit, except that $T_a$ appears in the denominator of the Carnot efficiency rather than $T_s$, an aspect that has previously been derived in the context of a "dissipative" heat engine (Renno and Ingersoll, 1996; Bister and Emanuel, 1998). Note that in the presence of positive heat storage changes, as is the case during the day, the maximum power that can be derived from the heat flux $J_{in}$ is reduced. That is, the increase in heat storage within the engine ($dU_e/dt > 0$) results in a lower efficiency in converting heat into power (with the efficiency given by the ratio $G/J_{in}$), consistent with our explanation in the Introduction of why we refer to this effect as that of a cold heat engine.

## 2.2 Energy balance constraints

We next use the energy balance constraints of the surface and the whole system to express $dU_e/dt$ and $T_s - T_a$ in terms of the absorption of solar radiation at the surface, $R_s$, and the turbulent heat flux $J_{in}$. This will allow us to replace these two terms in Eq. (4), so that the power $G$ only depends on $R_s$ and $J_{in}$. Note that we refer to the atmospheric heat storage change, $dU_a/dt$ in the following rather than the engine heat storage change, $dU_e/dt$. The difference is that when we apply the thermodynamic limit on the atmosphere, the heat storage is also affected by the net exchange of longwave radiation, which adds another term to the energy and entropy budget but which does not go through the engine as a heat flux. However, the resulting limit remains unaffected, as shown in Appendix A.

The surface energy balance constrains the relationship between the heat flux $J_{in}$ and the temperature difference that drives the heat engine, $T_s - T_a$. We express this balance by

$$R_s - k(T_s - T_a) - J_{in} - \frac{dU_s}{dt} = 0, \qquad (5)$$

where we linearize the net longwave radiative exchange, $R_{l,net} = k(T_s - T_a)$, between the surface and the atmosphere

**Table 1.** Variables and parameters used in this study.

| Symbol | Variable | Units | Use or assumption |
|--------|----------|-------|-------------------|
| $D$ | Frictional dissipation | $\mathrm{W\,m^{-2}}$ | Assumed to be in steady state, with $D = G$ |
| $G$ | Convective power | $\mathrm{W\,m^{-2}}$ | Eqs. (1), (3) and (4) |
| $J_{\mathrm{in}}$ | Turbulent fluxes of sensible and latent heat | $\mathrm{W\,m^{-2}}$ | Eqs. (1), (2) and (5) |
| $J_{\mathrm{opt}}$ | Turbulent fluxes $J_{\mathrm{in}}$ optimized to yield max. power | $\mathrm{W\,m^{-2}}$ | Eq. (7) |
| $J_{\mathrm{out}}$ | Cooling rate of the heat engine | $\mathrm{W\,m^{-2}}$ | Eqs. (1) and (2) |
| $k$ | Radiative parameterization constant | $\mathrm{W\,m^{-2}\,K^{-1}}$ | Used in linearization of $R_{\mathrm{l,net}}$ |
| $R_{\mathrm{l,out}}$ | Flux of terrestrial radiation to space | $\mathrm{W\,m^{-2}}$ | Assumed to be in steady state, with $R_{\mathrm{l,out}} = R_{\mathrm{s,avg}}$ |
| $R_{\mathrm{s}}$ | Surface absorption of solar radiation | $\mathrm{W\,m^{-2}}$ | Forcing |
| $R_{\mathrm{s,avg}}$ | Surface absorption of solar radiation (average) | $\mathrm{W\,m^{-2}}$ | Eq. (6) |
| $T_{\mathrm{a}}$ | Atmospheric temperature | K | Assumed to be the radiative temperature |
| $T_{\mathrm{e}}$ | Temperature of the heat engine | K | Assumed to be similar to the surface temperature |
| $T_{\mathrm{s}}$ | Surface temperature | K | – |
| $\mathrm{d}U_{\mathrm{a}}/\mathrm{d}t$ | Change in atmospheric heat storage | $\mathrm{W\,m^{-2}}$ | Eq. (6) |
| $\mathrm{d}U_{\mathrm{e}}/\mathrm{d}t$ | Change in heat storage within heat engine (assumed to be the same as $\mathrm{d}U_{\mathrm{a}}/\mathrm{d}t$ in Sect. 2.2) | $\mathrm{W\,m^{-2}}$ | Eqs. (1)–(4) |
| $\mathrm{d}U_{\mathrm{s}}/\mathrm{d}t$ | Change in ground heat storage (or ground heat flux) | $\mathrm{W\,m^{-2}}$ | Prescribed from observations, Eq. (6) |
| $\mathrm{d}U_{\mathrm{tot}}/\mathrm{d}t$ | Change in total heat storage | $\mathrm{W\,m^{-2}}$ | Eq. (6) |

and where $\mathrm{d}U_{\mathrm{s}}/\mathrm{d}t$ describes heat storage changes below the surface, which is represented by the ground heat flux. This formulation of the surface energy balance can be used to express the temperature difference, $T_{\mathrm{s}} - T_{\mathrm{a}}$, as a function of $R_{\mathrm{s}}$, $J_{\mathrm{in}}$, and heat storage changes below the surface, $\mathrm{d}U_{\mathrm{s}}/\mathrm{d}t$.

The energy balance of the whole system, neglecting heat advection terms, yields a constraint of the form

$$\frac{\mathrm{d}U_{\mathrm{tot}}}{\mathrm{d}t} = \frac{\mathrm{d}U_{\mathrm{a}}}{\mathrm{d}t} + \frac{\mathrm{d}U_{\mathrm{s}}}{\mathrm{d}t} = R_{\mathrm{s}} - R_{\mathrm{l,out}} = R_{\mathrm{s}} - R_{\mathrm{s,avg}}, \quad (6)$$

where $\mathrm{d}U_{\mathrm{tot}}/\mathrm{d}t$ is the total change in heat storage within the surface–atmosphere system. We assume this balance to be in a steady state when averaged over day and night, so that on average, $R_{\mathrm{l,out}} = R_{\mathrm{s,avg}}$, where $R_{\mathrm{s,avg}}$ is the temporal mean of $R_{\mathrm{s}}$ taken over the whole day. The energy balance of the whole system provides an expression for $\mathrm{d}U_{\mathrm{a}}/\mathrm{d}t$ as a function of the instantaneous value of absorbed solar radiation, $R_{\mathrm{s}}$, the mean absorption of solar radiation, $R_{\mathrm{s,avg}}$, and the ground heat flux, $\mathrm{d}U_{\mathrm{s}}/\mathrm{d}t$.

## 2.3 Maximization of convective power

The surface energy balance (Eq. 5) can now be used to express the temperature difference that drives the heat engine, $T_{\mathrm{s}} - T_{\mathrm{a}}$, in the thermodynamic limit given by Eq. (4), while the energy balance of the whole system (Eq. 6) can be used to constrain the terms describing the changes in heat storage, $\mathrm{d}U_{\mathrm{a}}/\mathrm{d}t$. As the power $G$ is an increasing function of $J_{\mathrm{in}}$, but the temperature difference declines with greater values of $J_{\mathrm{in}}$, the power has a maximum, which is referred to as the maximum power limit. This limit can be derived analytically by $\partial G / \partial J_{\mathrm{in}} = 0$ and is associated with an optimum heat flux of

the form

$$J_{\mathrm{opt}} \approx \frac{1}{2}\left( R_{\mathrm{s}} - \frac{\mathrm{d}U_{\mathrm{s}}}{\mathrm{d}t} + \frac{\mathrm{d}U_{\mathrm{a}}}{\mathrm{d}t} \right). \quad (7)$$

This expression is consistent with previous work where the optimum heat flux is given by $J_{\mathrm{opt}} = R_{\mathrm{s}}/2$ in the absence of heat storage changes (Kleidon and Renner, 2013a, b). It is, however, modulated by heat storage changes, and it matters whether these changes take place below the surface or in the lower atmosphere as the two forms of heat storage change enter Eq. (7) with a different sign.

We next consider the two limiting cases. The first limit is when the heat storage changes take place primarily below the surface, like an open water surface of a lake. In this case, $\mathrm{d}U_{\mathrm{s}}/\mathrm{d}t \approx \mathrm{d}U_{\mathrm{tot}}/\mathrm{d}t$ (and $\mathrm{d}U_{\mathrm{a}}/\mathrm{d}t \approx 0$), and the optimum heat flux reduces to

$$J_{\mathrm{opt}} \approx \frac{R_{\mathrm{s,avg}}}{2}. \quad (8)$$

The other limiting case is when the heat storage changes take place above the surface. Then, $\mathrm{d}U_{\mathrm{a}}/\mathrm{d}t \approx \mathrm{d}U_{\mathrm{tot}}/\mathrm{d}t$ (with $\mathrm{d}U_{\mathrm{s}}/\mathrm{d}t \approx 0$), and the optimum heat flux is

$$J_{\mathrm{opt}} \approx R_{\mathrm{s}} - \frac{R_{\mathrm{s,avg}}}{2}. \quad (9)$$

This expression implies that the optimum value of the turbulent heat flux varies directly with the absorbed solar radiation, $R_{\mathrm{s}}$, but has a constant offset given by half of the mean absorption, $R_{\mathrm{s,avg}}/2$. This offset should be a comparatively small value of about 80–100 $\mathrm{W\,m^{-2}}$, given a global mean value of surface absorption of solar radiation of 165 $\mathrm{W\,m^{-2}}$

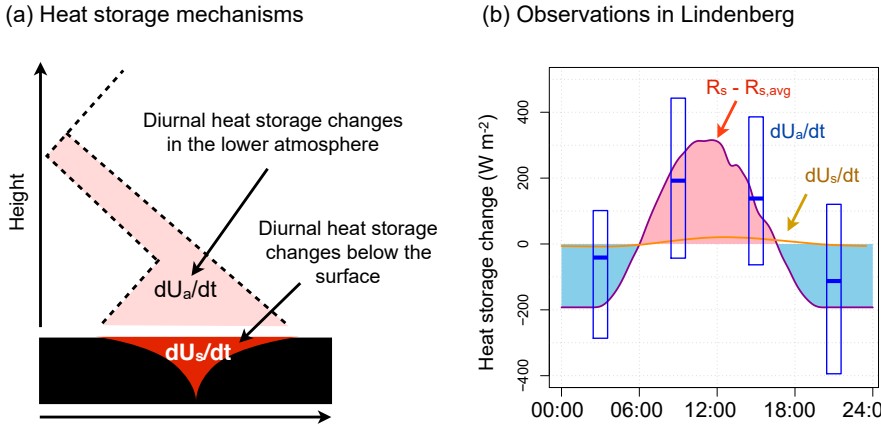

**Figure 2.** Diurnal changes in heat storage are reflected in variations of soil temperature near the surface and in variations of air temperature and humidity in the lower atmosphere. Panel **(a)** shows a schematic diagram of these heat storage changes. It shows a typical, colder nighttime profile with an inversion near the surface and a warmer daytime profile. The difference between the extremes of these temperature (and humidity) profiles (area shaded in light red) corresponds to the change in diurnal heat storage change in the lower atmosphere, $dU_a/dt$. Typical changes in belowground temperature profiles are also shown, with the heat storage change $dU_s/dt$ being marked in dark red. Panel **(b)** shows observations from Lindenberg, Germany, for the mean diurnal variation of absorbed solar radiation (shifted by its mean), $R_s - R_{s,avg}$, averaged for the month of June over the years 2006–2009 (red line, $n = 480$), the diurnal variation in heat storage in the lower atmosphere derived from 6-hourly radio soundings, $dU_a/dt$ (blue boxes represent the interquartile range and the horizontal thick blue line the median) and the ground heat flux, $dU_s/dt$ (orange line).

(Stephens et al., 2012). Note that the power, however, does not differ between the two cases and yields the same value of $G_{max} = (R_{s,avg}/2) \cdot (T_s - T_a)/T_a$.

Hence, the information on absorbed solar radiation (and the ground heat flux to account for $dU_s/dt$) is sufficient to estimate surface energy balance partitioning from the thermodynamic limit of maximum power.

## 2.4 Evaluation of the approach

Evaluating our estimate requires observations of absorbed solar radiation during the day, $R_s$, and the ground heat flux, $dU_s/dt$. From the diurnal course of $R_s$, the mean value of $R_{s,avg}$ can be calculated, which in turn yields an estimate for $dU_{tot}/dt$. Taken together with the ground heat flux, this yields the value of $dU_a/dt$, so that all terms in Eq. (7) can be specified. The resulting estimate of $J_{opt}$ can then be compared to observations of the turbulent heat fluxes or to the available energy, i.e., net radiation reduced by the ground heat flux.

## 3 Data sources

We use two types of data sources to test our approach. To test how reasonable the estimates are for the diurnal heat storage changes in the lower atmosphere, we first use 6-hourly radiosonde data from the DWD meteorological observatory Lindenberg in Brandenburg, Germany (data available at http://weather.uwyo.edu/upperair/sounding.html, last

access: 16 April 2018). These observations allow us to derive an estimate of the diurnal variations in temperature (and moisture) in the lower atmosphere and thus of $dU_a/dt$ (Fig. 2a). We use data from this site because this observatory provides a long and consistent record of four vertical profiles a day as well as surface energy balance components, while typically only two vertical profiles a day are taken during CE3 routine radiosonde measurements. We use observations from June for the years 2006 to 2009 and calculate the moist static energy at each 6 h interval and then take the difference over the time interval to obtain estimates for changes in atmospheric heat storage. These differences are then compared to the change in atmospheric heat storage expected from solar radiation, as described by Eq. (6).

We then use observations of absorbed solar radiation ($R_s$) and the ground heat flux ($dU_s/dt$) at six field sites in highly contrasting climatological settings (listed in Table 2) to calculate the turbulent heat fluxes from maximum power (Eq. 7). The six sites include a grassland and a forested site at Lindenberg, Brandenburg, Germany (Beyrich et al., 2006); three AmeriFlux sites (a tundra site at Anaktuvuk River, Alaska (Rocha and Shaver, 2011); a grassland site at Southern Great Plains, Oklahoma (Fischer et al., 2007; Raz-Yaseef et al., 2015); and a tropical rain forest site at Tapajos National Park, Brazil, (Goulden et al., 2004)); and a site in a planted pine forest at Yatir Forest in Israel (Rotenberg and Yakir, 2010, 2011). For each site, we use 1 month of observations for a summer period in which solar radiative heat-

**Table 2.** Site description of the six sites used for evaluating the estimations of the maximum power limit (with the letters referring to the graphs shown in Fig. 3). Also shown are the correlation statistics of the comparison to observations. The adjusted squared explained variance of the linear regression of $J_{opt}$ to observed net radiation ($R_n = R_s - R_{l,net}$) minus ground heat flux $R_n - dU_s/dt$ is reported as $r^2$. Standard error of slope and intercept of the regression are derived by a pre-whitening procedure to reduce the effect of serial correlation of the residuals (Newey and West, 1994; Zeileis, 2004).

| Site | Description | $r^2$ | Slope | Intercept | Reference |
|------|-------------|-------|-------|-----------|-----------|
| A | Tundra (open shrubland), USA Anaktuvuk River (unburned site) 68°56′ N, 150°16′ W Data used for June 2009, $n = 1392$ | 0.972 | 1.138 ±0.019 | −54.80 ±5.17 | Rocha and Shaver (2011) https://doi.org/10.17190/AMF/1246144 |
| B | Cropland, USA ARM Southern Great Plains site 36°36′ N, 97°29′ W Data used for June 2009, $n = 1060$ | 0.993 | 1.106 ±0.015 | −73.30 ±5.04 | Fischer et al. (2007), Raz-Yaseef et al. (2015) https://doi.org/10.17190/AMF/1246027 |
| C | Temperate grassland, Germany DWD Falkenberg boundary layer site 52°10′ N, 14°7′ E Data used for June 2009, $n = 1440$ | 0.982 | 1.099 ±0.012 | −36.38 ±3.56 | Beyrich et al. (2006) |
| D | Pine forest, Germany DWD Falkenberg boundary layer site 52°11′ N, 13°57′ E Data used for June 2009, $n = 1438$ | 0.982 | 1.023 ±0.011 | −37.87 ±4.34 | Beyrich et al. (2006) |
| E | Pine forest, Israel Yatir Forest 31°20′ N, 35°3′ E Data used for June 2006, $n = 1440$ | 0.998 | 1.086 ±0.006 | −53.87 ±2.22 | Rotenberg and Yakir (2010, 2011) |
| F | Tropical rain forest, Brazil Santarem km83 logged forest 3°1′ S, 54°58′ W Data used for June 2002, $n = 1053$ | 0.999 | 0.995 ±0.003 | −59.82 ±1.23 | Goulden et al. (2004) https://doi.org/10.17190/AMF/1245995 |

ing of the surface is highest and the effects of heat advection are minor; we estimate turbulent fluxes associated with maximum power (using Eq. 7) and compare these to the observed fluxes.

## 4 Results

We first evaluate the extent to which diurnal variations in solar radiation are buffered by heat storage changes in the lower atmosphere. To do so, we use the diagnosed variations of moist static energy from the radiosoundings in Lindenberg, Germany and compare these to the mean variation in absorbed solar radiation at the surface as well as variations in the ground heat flux at the site in Fig. 2b. The comparison shows that the heat storage variations in the lower atmosphere are substantially greater than the ground heat flux so that the diurnal variations in solar radiation are mostly buffered by the lower atmosphere. Although there is considerable variation (as indicated by the blue boxes), mostly due to pressure changes and advective effects, these varia-

tions follow the temporal course of what is expected from the variation in absorbed solar radiation (as described by Eq. 6). This confirms our conjecture that the diurnal variations in solar radiation on land are buffered primarily by heat storage changes in the lower atmosphere. This buffering of the diurnal variations over the land surface is rather different to how an open water surface buffers these variations (as also shown by observations; Liu et al., 2009; this is an aspect used previously to explain the difference in climate sensitivity of land and ocean surfaces; Kleidon and Renner, 2017).

The comparison of the estimated surface energy balance partitioning from maximum power to observations at the six sites is shown in Fig. 3. The correlations are summarized in Table 2 in terms of the correlation coefficient as well as the slope and intercept. During nighttime, there is a mismatch between our approach and observations, which is represented by the intercept shown in Table 2. This mismatch may be explained by the prevalent stable nighttime conditions in which the atmosphere does not act as a heat engine, an aspect that we did not consider in our approach. During daytime, we find very high correlations of above 95 % between the estimated

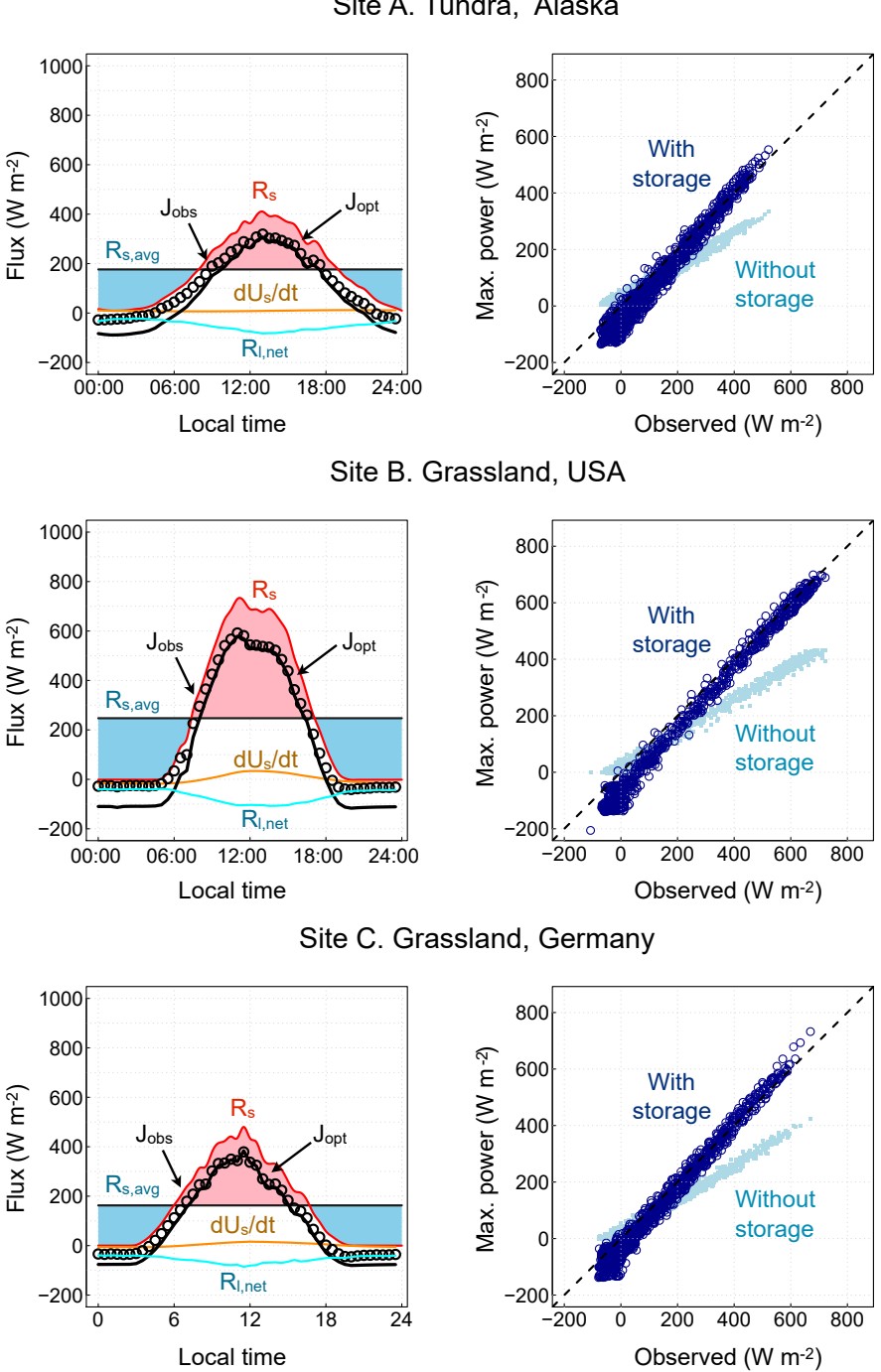

**Figure 3.**

turbulent fluxes from the maximum power limit with observed net radiation (reduced by the ground heat flux), with a very good match of the estimated slopes in the correlation within 15 % of the observed. This high level of agreement is found across the range of climatological settings shown in Fig. 3.

Also note that the maximum power limit without an explicit consideration of heat storage changes (i.e., with $dU_s/dt = 0$ and $dU_a/dt = 0$ in Eq. (7), as in Kleidon et al., 2014, and as indicated by light blue points in Fig. 3) estimates turbulent fluxes that also result in a high correlation but with a magnitude that is too low compared to observa-

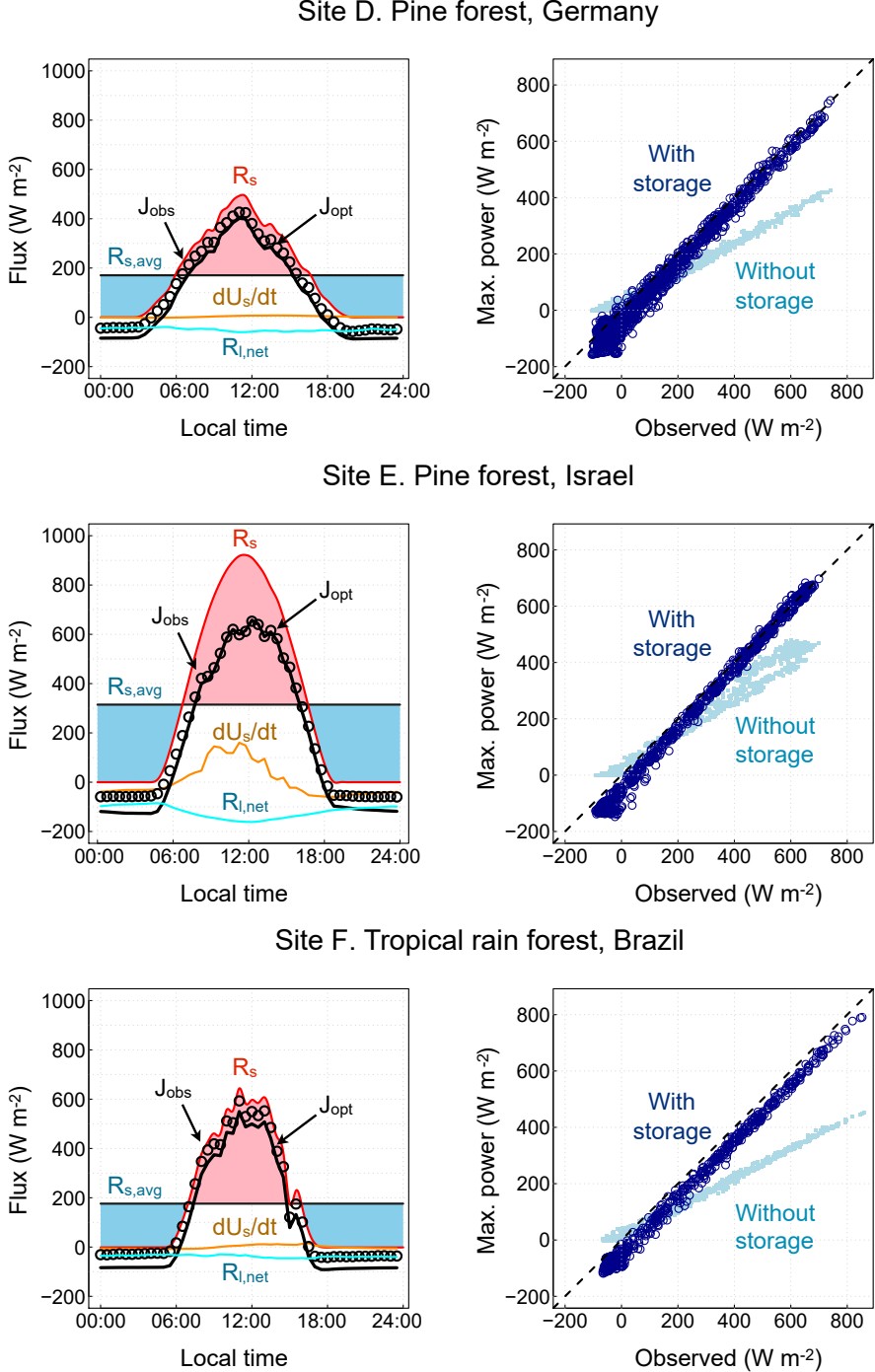

**Figure 3.** Mean diurnal cycle of the absorption of solar radiation at the surface ($R_s$, red line, observed), ground heat flux ($dU_s/dt$, orange line, observed), and turbulent heat fluxes estimated by maximum power ($J_{opt}$, black line, estimated) and observations ($J_{obs}$, black circles, observed) for a selected month for six field observations in **(a)** a tundra ecosystem in Alaska, **(b)** a cropland in the midwestern US, **(c, d)** a grassland and pine forest in a temperate environment in Germany, **(e)** a planted pine forest in an arid environment in Israel, and **(f)** a tropical rain forest in the humid Amazon Basin in Brazil. The comparison of the turbulent heat fluxes estimated from maximum power to energy balance measurements is shown for 30 min observations in the right panel for each site for two cases of thermodynamic limits that differ by their consideration of heat storage changes (dark blue: with storage, as in Eq. (4); light blue: without storage, i.e., $dU_s/dt = 0$ and $dU_a/dt = 0$ so that $J_{opt} = R_s/2$). More information on the sites and the correlation statistics are provided in Table 2.

tions. This high level of agreement of the maximum power limit with diurnal heat storage changes suggests that it is an adequate description of surface energy balance partitioning and land–atmosphere exchange at the diurnal timescale, so that turbulent fluxes appear to operate near their thermodynamic limit. It further shows that it is critical to account for diurnal variations in heat storage in the thermodynamic limit to adequately represent the magnitude of the observed turbulent fluxes.

## 5   Discussion

Our approach, of course, only represents a general description of the full dynamics of surface–atmosphere exchange. Notable effects not considered in our approach that could alter the results and potentially modulate the outcome of the maximum power limit include a more detailed representation of radiative transfer, a distinction between the sensible and latent heat fluxes which result in different forms of storage changes in the atmosphere, entrainment effects at the top of the boundary layer, advection and coupling to large-scale atmospheric processes, and a better representation of nighttime processes, particularly regarding the formation of stable conditions at night that prevent convection to occur. These aspects can be explored further in future extensions. Yet even at this highly simplified level, the agreement of the estimated flux partitioning with observations is rather remarkable, indicating that the dominant forcing and the dominant constraints are captured by our approach.

Our results emphasize the importance of considering the constraint imposed by the second law of thermodynamics on land–atmosphere exchange. While the complex, turbulent nature of this exchange makes it seem almost impossible to describe its outcome in simple terms, the generation of turbulent kinetic energy that drives the diurnal development of the convective boundary layer is nevertheless constrained by thermodynamics. The very good agreement of our results with observations suggests that this constraint imposed by thermodynamics is relevant to this generation, and land–atmosphere exchange appears to operate near this thermodynamic limit. This is consistent with previous research that applied thermodynamics and/or heat engine frameworks to atmospheric motion, for instance approaches using the proposed principle of maximum entropy production (Paltridge, 1978; Ozawa and Ohmura, 1997; Lorenz et al., 2001; Ozawa et al., 2003) or applications to hurricanes and atmospheric convection (Emanuel, 1999; Pauluis and Held, 2002a, b). Note that our maximization of power is almost identical to the maximization of material entropy production, as we assume a steady state in which power equals dissipation ($G = D$), and entropy production by turbulence is then given by $D/T$, where $T$ is the temperature at which dissipation occurred (with $T \approx T_s$). Yet our approach differs in that it specifically considered the effect of heat stor-

age changes in altering the thermodynamic limit and feedbacks with the surface energy balance that altered the driving temperature difference of the heat engine. The heat storage changes in the lower atmosphere result in an additional term in the Carnot limit, and this can explain why the land–atmosphere system functions quite differently with its pronounced diurnal variations in turbulent fluxes than the temporally much more uniform turbulent fluxes over open water surfaces (e.g., Liu et al., 2009; Kleidon, 2016; Kleidon and Renner, 2017). Thermodynamics combined with these two additional factors then provide sufficient constraints on the magnitude of turbulent heat fluxes. It would seem that this could provide valuable information to better parameterize turbulent fluxes within the Monin–Obukhov similarity theory for unstable conditions, specifically regarding the stability functions that are used in this approach (e.g., as in Louis, 1979).

This insight that surface energy balance partitioning is predominantly determined by the local partitioning of the absorbed solar radiation is rather different than the way this exchange is commonly represented in climate models. In these models, surface exchange is parameterized using the aerodynamic bulk approach, in which the aerodynamic drag of the surface and horizontal wind speeds play a dominant role that is modulated by stability functions. Our approach differs in that solar radiation plays the dominant role in surface exchange by the local generation of buoyancy and power to drive convection, rather than wind speed and aerodynamic roughness as what the bulk method would suggest. A recent intercomparison between a number of commonly used land surface models (Best et al., 2015) shows, however, that land surface models using the bulk method generally underestimate the strong correlation of turbulent fluxes with downward solar radiation found in observations. Our approach can resolve this bias and suggests that the bulk method may underestimate the effect of the local forcing by solar radiation on surface–atmosphere exchange.

We think that our approach provides ample opportunities for future applications and research. First, the simple expression of how turbulent heat fluxes on land vary during the day, as given by Eq. (9), provides an easy way to get a first-order estimate. It could serve as a baseline estimate that is solely based on physical principles, specifically, the first and second law of thermodynamics, and does not require tuning. This expression should nevertheless be further evaluated in a broader range of climatological conditions and over extended time periods to identify possible shortcomings, for instance with respect to the simple parameterization of longwave radiation or regarding the omission of advective effects. For a broader range of applicability, the approach would need to be extended further to derive an expression for near-surface air temperature, which would be related to the changes in atmospheric heat storage ($dU_a/dt$), for the aerodynamic conductance, and for boundary layer development, and the turbulent heat fluxes should be separated into the fluxes of sensible and

latent heat. It would also be instructive to compare the power associated with the limit with estimates of the turbulent kinetic energy generation rate from observations to develop another possibility for testing the maximization approach.

Our approach can then be used to evaluate aspects of global change analytically, such as land cover change or global warming, providing an alternative approach to these topics that complements complex, numerical modeling approaches. More generally, the success of our approach in reproducing observations very well constitutes another example of processes in complex systems appearing to evolve to and operate at their thermodynamic limit (Ozawa et al., 2003; Martyushev and Seleznev, 2006; Kleidon et al., 2010; Kleidon, 2016). This, in turn, encourages the application of thermodynamics to a broader range of questions and topics to understand the evolution and emergent dynamics of complex Earth systems.

## 6   Conclusions

We formulated a Carnot limit which accounts for heat storage changes within the atmospheric heat engine and used this limit to estimate the partitioning of the solar radiative forcing into radiative and turbulent cooling at the diurnal timescale. In contrast to common approaches to describe near-surface turbulent heat transfer into the atmosphere, we explicitly consider the thermodynamic constraint imposed by the second law of thermodynamics by treating turbulent heat fluxes and convection as the result of a heat engine. The maximization of the work output of this convective heat engine then yields estimates of turbulent fluxes that compare very well to observations across a range of climates and do not require empirical parameterizations. This demonstrates that our approach represents an adequate, general description of the land surface energy balance that only uses physical concepts and that does not rely on semiempirical turbulence parameterizations.

We conclude that turbulent fluxes over land appear to operate near its thermodynamic limit by which the power of the convective heat engine is maximized. This limit is shaped by the second law of thermodynamics, as in the case of the Carnot limit of a heat engine in classical thermodynamics, but also requires the consideration of two additional factors that relate the heat engine to its environmental setting. The first factor relates to the strong diurnal variation of solar radiation, which results in diurnal heat storage changes. Over land these changes are buffered primarily in the lower atmosphere and these modulate the Carnot limit, resulting in a reduced efficiency and in what we referred to as a cold heat engine. Second, the limit of maximum power of the atmospheric heat engine is shaped by the trade-off in the driving temperature difference between surface and atmosphere, which decreases with greater turbulent heat fluxes. This trade-off results in the maximum power limit and represents a strong coupling between surface conditions and the lower atmosphere.

Overall, our study shows that thermodynamics adds a highly relevant constraint to land–atmosphere coupling. This thermodynamic approach to the surface energy balance and land–atmosphere interactions should help us to better understand the role of the land surface and terrestrial vegetation in the climate system and how they interact with global change.

**Data availability.** All data used in this study are available upon request by contacting the correspondence author. TS3

## Appendix A: Effects of radiative exchange on the limit of a cold heat engine

The derivation of the Carnot limit with heat storage changes in Sect. 2.1 assumed in the first law that the heat storage change within the heat engine is entirely caused by the heat flux $J_{in}$. When applying this approach to turbulent fluxes between the land surface and the atmosphere, one also needs to consider the net transport of energy by radiative exchange between the surface and the atmosphere. In the derivation above, this net exchange is represented by the flux $R_{l,net}$. This flux contributes to the heat storage change in the lower atmosphere, but it is not driven by the heat engine. This results in a small inconsistency when applying the limit of Sect. 2.1 to the lower atmosphere. In the following, we show that the limit derived in Sect. 2.1 is still valid. However, whether the lower atmosphere is opaque to longwave radiative transfer and absorbs $R_{l,net}$ or whether it is instead transparent makes a difference in the justification, which is why we included this derivation here rather than in the main text.

In the following, we assume that the radiative–convective layer of the lower atmosphere is sufficiently opaque and absorbs the net longwave radiation of the surface, $R_{l,net}$. Then, the first law described by Eq. (1) becomes the energy balance of the lower atmosphere and changes to

$$\frac{dU_a}{dt} = J_{in} + R_{l,net} - R_{l,out} - G + D, \tag{A1}$$

where $G = D$ and $R_{l,out} = R_{s,avg}$ in steady state.

The second law (Eq. 2) obtains another term related to the entropy being added by the warming due to the absorption of the net flux of longwave radiation, $R_{l,net}$. As this warming takes place at the prevailing physical temperature of the atmosphere (rather than the potential temperature), its temperature is likely closer to $T_a$ rather than $T_e$ or $T_s$. Hence, the entropy budget changes to

$$\frac{1}{T_e}\frac{dU_a}{dt} = \frac{J_{in}}{T_s} + \frac{D}{T_e} - \frac{R_{l,out}}{T_a} + \frac{R_{l,net}}{T_a} + \sigma_{irr}. \tag{A2}$$

As in Sect. 2.1, we can combine Eqs. (A1) and (A2), solve them for $D$ ($= G$), and obtain a limit on the power ($G$) by assuming that the entropy production $\sigma_{irr} = 0$:

$$G = J_{in} \cdot \frac{T_e}{T_s} \cdot \frac{T_s - T_a}{T_a} - \frac{dU_e}{dt} \cdot \frac{T_e - T_a}{T_a}. \tag{A3}$$

This is the same expression as Eq. (3), so that the effect of net longwave radiative transfer actually cancels out.

In the case in which the lower atmosphere is comparatively transparent for longwave radiation, the flux $R_{l,net}$ passes through the lower atmosphere without being absorbed. In this case, Eqs. (1) to (4) remain unaffected.

**Author contributions.** AK and MR jointly developed the idea for this paper. AK performed the theoretical derivation and MR the data analysis. AK led the writing of the manuscript with input from MR.

**Competing interests.** The authors declare that they have no conflict of interest.

**Special issue statement.** This article is part of the special issue "Thermodynamics and optimality in the Earth system and its subsystems (ESD/HESS inter-journal SI)". It is not associated with a conference.

**Acknowledgements.** We thank two anonymous reviewers for their helpful reviews and Henk de Bruin, Andreas Chlond, Pierre Gentine and Aljosa Slamersak for constructive discussions on land–atmosphere exchange. This research contributes to the "Catchments As Organized Systems (CAOS)" research group (FOR 1598) funded by the German Science Foundation (DFG). We acknowledge the University of Wyoming for making the radio sounding data available at http://weather.uwyo.edu/upperair/sounding.html (last access: 16 April 2018). Data from Site A were funded by NSF grant 1556772 to the University of Notre Dame. Data from Site B were supported by the Office of Biological and Environmental Research of the US Department of Energy under contract no. DE-AC02-05CH11231 as part of the Atmospheric Radiation Measurement Program, ARM). Data for Site C and D were provided by the Deutscher Wetterdienst (DWD) –Meteorologisches Observatorium Lindenberg/Richard-Assmann Observatorium. They were obtained in the context of the Coordinated Energy and Water Cycle Observation Project (CEOP), which was initiated as an international effort in 1998 by the World Climate Research Programme (WCRP) Global Energy and Water Cycle Experiment (GEWEX) Hydrometeorology Panel (GHP) in support of global climate research interests. Data for Site E were kindly provided by Eyal Rotenberg and Daniel Yakir. Data for Site F were provided by the AmeriFlux data server: http://ameriflux.ornl.gov (last access: 16 April 2018).

The article processing charges for this open-access publication were covered by the Max Planck Society.

Edited by: Michel Crucifix
Reviewed by: two anonymous referees

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

## Remarks from the language copy-editor

## Remarks from the typesetter