# Peer review of "Diurnal land surface energy balance partitioning estimated from the thermodynamic limit of a cold heat engine"

_Earth System Dynamics, 2018_

## Referee Comment (RC1) · Anonymous Referee #1 · 23 May 2018

**General comments**

This paper investigates diurnal variations of turbulent heat fluxes and energy partitioning at the surface of the earth. The turbulent motion is assumed to produce a maximum rate of mechanical work by convective heat transport from the surface to the atmosphere through a Carnot-like cycle, and the surface temperature is assumed to be determined by the energy balance conditions in the radiation field. A key process assumed in this study is that part of the turbulent heat flux is stored in the heat engine, and this storage does not affect the atmospheric temperature (Ta) and the radiation thereof (k Ta). Under such assumptions, the turbulent heat flux at the surface can

be estimated to be a function of solar radiation (Rs), its daily mean (Rs,avg) and the ground heat flux (dUs/dt), and the estimated heat flux shows remarkable agreement with observations (Figure 3). This reviewer finds many parts of this paper interesting and provocative, and recommends an immediate publication of this paper. However, I have a few comments and suggestions that will be helpful to improve the quality and interpretation of the results before the publication of this paper.

Specific comments:

**1. A "cold" heat engine?**

The authors refer to the atmospheric engine with heat storage as a "cold" heat engine because of its lower efficiency compared to a usual heat engine without such storage (page 5, line 25). I have a different opinion on this point because the actual power of this engine is not very low for the following reason.

It is true that an increase in heat storage in the engine (dUa/dt > 0) results in lower power generation because the heat stored in the engine (Te  $\approx$  Ts) is not transferred to the atmosphere with lower temperature (Ta), and thus cannot produce work according to the second law. This heat storage increase therefore results in a reduction in the power generation rate in Eq. (4). However, for the same reason of this heat storage, the reduction in the surface temperature against Jin is reduced when there is such heat storage increase (dUa/dt > 0). Thus, the optimum-state surface heat flux (Jopt) becomes larger when dUa/dt > 0, as is shown in Eq. (7). When we combine these two effects (negative and positive effects in the heat flux) together, and substitute Jopt (7) into (4), we get the optimum-state power generation rate, with the total energy balance condition (6), as

Gopt = Rs,avg/2 (Ts - Ta)/Ta. — (a)

This means that the large variation in Rs is cancelled out by the buffering effect of the heat storage, and Gopt is nearly constant over time, which is indeed identical to the

steady-state generation rate found by Kleidon and Renner (2013). So the heat storage seems to act to regulate the rapid change in Rs (the driving force) to realize a nearly constant rate of power generation by storing heat in daytime and discharging heat in nighttime. I would therefore suspect a "buffering" effect of the heat engine with the heat storage rather than a "cold" heat engine. It may also be interesting to see if Gopt of the atmospheric system really shows such regulation by using the observed data (Rs, dUs/dt and Ts).

2. Figure 3 and its interpretation.

One of the most striking results of this paper is that the predicted Jopt shows remarkable agreement with the observed Jobs (Fig. 3). Also, one can see that Jopt (and Jobs) is generally smaller than Rs by a nearly constant value of about 50-100 W/m2. This result can be explained directly from Eqs. (6) and (7) as

Jopt = Rs - Rs,avg/2 - dUs/dt  $\approx$  Rs - Rs,avg/2. — (b)

The last approximation (dUs/dt  $\approx$  0) is written in the text as a limiting case (page 6, line 26). Eq. (b) clearly shows that the optimum-state turbulent heat flux should be Rs minus a constant value (Rs,avg/2  $\approx$  80-100 W/m2) for the most of the observation sites (Fig. 3A-D, F) where dUs/dt is nearly negligible. I think Eq. (b) captures valuable information on the turbulent heat flux at the optimum (maximum power) state, and may be explained in the results or discussions of this paper. The readers of this paper will then be able to use it as a diagnostic tool for future investigations on turbulent heat fluxes obtained from observations and GCM simulations.

I am somewhat skeptical about the region where Rs  $\leq$  Rs,avg/2 and Jopt becomes negative (Fig. 3). This region implies a situation of "inverse" heat flux from the atmosphere to the surface, and convective motion as well as convective heat flux cannot occur in this situation. The validity of the assumptions used in this study becomes questionable (even G and D become negative) under such situation. I think this can be a cause of the failure in predicting Jobs in this situation, in addition to the effect of the

СЗ

prevalent stable nighttime stratification in the boundary layer (page 8, line 6).

Technical corrections:

1. Page 1, line 21: "in by biases".

Just typo.

2. Figure 2

I cannot see what is the meaning of the rectangular boxes (blue) in Fig. 2B. Perhaps a range of standard deviation? It may be good to explain this in the caption. Also, the "minus" sign in "Rs - Rs,avg" should not be in the lowercase.

3. Figure 3

This figure should be made larger so that one can see the details of the results. Also, some of the arrows of Jobs and Jopt are not properly located for the corresponding lines and circles. It seems good to adjust them.

---

## Referee Comment (RC2) · Anonymous Referee #2 · 5 Jul 2018

General comments

This is one of a series of papers using thermodynamic principles (first and second law) together with optimization concepts to investigate aspects of the climate system. Here, the authors apply this approach to turbulent energy fluxes at land surfaces. They show that fluxes derived from optimizing a Carnot cycle modified by heat storage compare well with observed fluxes, and conclude that the applied concept can help to better understand the role of land surfaces and to parameterize the surface-atmosphere interaction. I find the paper interesting as it illustrates a promising approach, and may help to stimulate further investigations in this direction. Thus, I would recommend pub-

lication. However, I have few points, which, in my view, need clarification/consideration before final acceptance.

Specific comments

1) 'cold heat engine': The authors frequently use the term 'cold heat engine', which, honestly, was not known to me before. It seems that a cold heat engine is defined as a heat engine with some storage (P2L29), but a more precise definition may be given.

2) Fig. 1 and Eqs. 1,5,6 : From Fig. 1 it seems that the heat engine discussed by the authors is confined to the radiative-convective layer with Jout being a flux into the free atmosphere above. However, combining Eqs 1,4,5 gives Jout=Rl,out-Rl,net, i.e. the cooling of the whole atmospheric column by thermal radiation. Thus, either it is assumed that there is no exchange between the radiative-convective layer and the free atmosphere, or the heat engine comprises the whole column. This needs to be clarified (in Fig.1 and/or the text introducing the heat engine).

3) Eq. 2: Eq. 2 gives the entropy budget of the heat engine. However, Jout=Rl,out-Rl,net (see above), and Rl,net is the sum of thermal flux coming from the atmosphere (approx. R,l,out, say) and from the surface (Rl,surf). Thus, instead of Jout/Ta I would expect a term (Rl,surf/Ts) and something like 2Rl,out/Ta appearing in Eq. 2, representing both the import of entropy from the soil and the respective export from the atmosphere. It seems that Rl,surf/Ts can be of the same order as Jin/Ts. The authors need to explain why the entropy import from the surface (Rl,surf/Ts) is not considered, in particular as Jout/Ta is used to obtain Eqs. 3,4,7.

4) Eq. 7 (Jopt vs Jin, Part I): Eq. 7 gives an estimate for Jin derived from optimization based on the second law. However, using Eqs. 1,5 to replace dUa/dt and dUs/dt in Eq.7 (or replacing dUs/dt in Eq. 5 by Jobt with dUa/dt as described in Sec. 2.4) shows (if I'm not wrong) that Jobt is not equal Jin. Thus, while Jobt results from utilizing the second law it seems not to be consistent with energy conservation (the first law) within the same model framework (Eqs 1,5). If the conclusion (and the approach taken) that

the turbulent fluxes optimize the heat engine constrained by energy conservation holds, this surprises me. What is the explanation (perhaps it is trivial)?

5) Jopt vs Jin, Part II: The difference between Jopt and Jin (as explained above) is given by Rl,net-Rl,out/2 (again, I hope that I'm not wrong). In Fig 3, although it is hard to judge, this difference seems to be relatively large, and larger than the difference between Jopt and Jobs. If so, this surprises me too. Perhaps, the authors may like to compute this Jin (consistent with energy conservation constraint), compare it with Jopt (Jobs), and discuss the result in the context of the optimization concept.

6) Fig2: I do not understand Fig 2a: A more comprehensive explanation may be given in the text: e.g. what defines the particular shape of the atmospheric heat storage change (pink area).

Technical corrections

1) P4L12: Rs -> Rs,ave

---

## Author Comment (AC1) · 5 Jul 2018

We thank the reviewer for the thoughtful and constructive review of our manuscript. In the following, we summarise the referee's comments in *italic*, provide our reply to each point, and suggest how we address these points in the revision. Note that the numbering is somewhat different to the referee's numbering to separate the referee's comments into separate points.

*Reviewer comment 1: The authors refer to the atmospheric engine with heat storage as a "cold" heat engine because of its lower efficiency compared to a usual heat engine without such storage (page 5, line 25). I have a different opinion on this point because*

[Figure]

*the actual power of this engine is not very low for the following reason. It is true that an increase in heat storage in the engine ($dU_a/dt > 0$) results in lower power generation because the heat stored in the engine ($T_e \approx T_s$) is not transferred to the atmosphere with lower temperature ($T_a$), and thus cannot produce work according to the second law. This heat storage increase therefore results in a reduction in the power generation rate in Eq. (4). However, for the same reason of this heat storage, the reduction in the surface temperature against Jin is reduced when there is such heat storage increase ($dU_a/dt > 0$). Thus, the optimum-state surface heat flux ($J_{opt}$) becomes larger when $dU_a/dt > 0$, as is shown in Eq. (7). When we combine these two effects (negative and positive effects in the heat flux) together, and substitute $J_{opt}$ (7) into (4), we get the optimum-state power generation rate, with the total energy balance condition (6), as*

$$G_{opt} = \frac{R_{s,avg}}{2} \cdot \frac{(T_s - T_a)}{T_a}. \tag{1}$$

*This means that the large variation in $R_s$ is cancelled out by the buffering effect of the heat storage, and $G_{opt}$ is nearly constant over time, which is indeed identical to the steady-state generation rate found by Kleidon and Renner (2013). So the heat storage seems to act to regulate the rapid change in $R_s$ (the driving force) to realize a nearly constant rate of power generation by storing heat in daytime and discharging heat in nighttime. I would therefore suspect a "buffering" effect of the heat engine with the heat storage rather than a "cold" heat engine.*

**Reply:** Thank you, these are very good and thoughtful points.

In terms of referring to our approach as a "cold" heat engine, this concerns the first term in the expression of power (Eq. 4), not the efficiency. With "cold", we do not refer to the comparison between heat storage changes taking place below or above the surface, but rather to the case where one uses the turbulent heat fluxes $J_{in}$ directly in the Carnot limit (i.e., $G = J_{in} \cdot (T_s - T_a)/T_a$), as one may naively do (and what we initially did, and which results in optimum heat fluxes that are notably too small, see light blue dots in Fig. 3).

Our motivation to refer to our Eq. 4 as the Carnot limit of a "cold" heat engine is similar to a cold car engine in winter. When the car engine is still cold in winter just after it has been started, one needs to hit the gas harder to get the same power. Our expression captures this effect: The heat flux needs to be larger to get a certain power, because the term $dU_a/dt$ reduces the effect of the heat flux on the power. As this is a storage effect similar to a cold car engine heating up, we think that the term "cold heat engine" nicely captures this storage effect.

This aspect of heat storage change alters the Carnot limit, as derived in sect. 2.1, and as such is independent of the feedbacks with surface temperature that come into play when this limit is evaluated in the surface-atmosphere system. So while the explanations that the maximum power limit actually results in the same limit of power, as nicely shown by the reviewer, we feel that the term "cold" heat engine is nevertheless appropriate, as we do not refer to the outcome in terms of maximising power, but rather to how the Carnot limit plays out in the context of heat storage changes.

In the revision, we will motivate and clarify the justification for the "cold" heat engine terminology in greater detail in the introduction and section 2.1, and will include some of the outcomes as pointed out by the reviewer in the discussion section.

***Reviewer comment 2:*** *It may also be interesting to see if $G_{opt}$ of the atmospheric system really shows such regulation by using the observed data ($R_s$, $dU_s/dt$ and $T_s$).*

**Reply:** Indeed, this is an excellent point. In principle, one should be able to diagnose this from the eddy flux data (i.e., the generation rate of turbulent kinetic energy), but this likely requires more data and information, and, quite frankly, we are not quite sure how this aspect should or can be evaluated.

We suggest to include this aspect in the revision as a motivation for future research in the discussion section.

***Reviewer comment 3:*** *One of the most striking results of this paper is that the pre-*

dicted $J_{opt}$ shows remarkable agreement with the observed $J_{obs}$ (Fig. 3). Also, one can see that $J_{opt}$ (and $J_{obs}$) is generally smaller than $R_s$ by a nearly constant value of about 50-100 W/m$^2$. This result can be explained directly from Eqs. (6) and (7) as

$$J_{opt} = R_s - \frac{R_{s,avg}}{2} - \frac{dU_s}{dt} \approx R_s - \frac{R_{s,avg}}{2}. \qquad (2)$$

The last approximation ($dU_s/dt \approx 0$) is written in the text as a limiting case (page 6, line 26). Eq. (2) clearly shows that the optimum-state turbulent heat flux should be $R_s$ minus a constant value ($R_{s,avg}/2 \approx 80 - 100$ W/m$^2$) for the most of the observation sites (Fig. 3A-D, F) where $dU_s/dt$ is nearly negligible. I think Eq. (2) captures valuable information on the turbulent heat flux at the optimum (maximum power) state, and may be explained in the results or discussions of this paper. The readers of this paper will then be able to use it as a diagnostic tool for future investigations on turbulent heat fluxes obtained from observations and GCM simulations.

**Reply:** Thanks for pointing this out. We agree that this expression is very useful and will add this to the results and discussion section in the revision.

**Reviewer comment 4:** *I am somewhat skeptical about the region where $R_s \leq R_{s,avg}/2$ and $J_{opt}$ becomes negative (Fig. 3). This region implies a situation of "inverse" heat flux from the atmosphere to the surface, and convective motion as well as convective heat flux cannot occur in this situation. The validity of the assumptions used in this study becomes questionable (even $G$ and $D$ become negative) under such situation. I think this can be a cause of the failure in predicting Jobs in this situation, in addition to the effect of the prevalent stable nighttime stratification in the boundary layer (page 8, line 6).*

**Reply:** Thank you very much for pointing this out. We fully agree and will mention and discuss this restriction in the revision.

**Reviewer comment 5:** *Page 1, line 21: "in by biases". Just typo.*

**Reply:** Thank you, we will correct this in the revision.
***Reviewer comment 6:*** *I cannot see what is the meaning of the rectangular boxes (blue) in Fig. 2B. Perhaps a range of standard deviation? It may be good to explain this in the caption. Also, the "minus" sign in "$R_s - R_{s,avg}$" should not be in the lowercase.*

**Reply:** The blue boxes mark the first and third quantile, with the horizontal bar showing the median of the analysed data. We will clarify this in the figure caption in the revision.

***Reviewer comment 7:*** *This figure should be made larger so that one can see the details of the results. Also, some of the arrows of $J_{obs}$ and $J_{opt}$ are not properly located for the corresponding lines and circles. It seems good to adjust them.*

**Reply:** Agreed. We will enlarge the figure and locate the labelling more properly in the revision.

―――――――――――――――――――

---

## Author Comment (AC2) · 5 Jul 2018

We thank the reviewer for the constructive and helpful review of our manuscript. In the following, we summarise the referee's comments in *italic*, provide our reply to each point, and suggest how we address these points in the revision.

*Reviewer comment 1: The authors frequently use the term "cold heat engine", which, honestly, was not known to me before. It seems that a cold heat engine is defined as a heat engine with some storage (P2L29), but a more precise definition may be given.*

**Reply:** We introduce this term here. To our knowledge it has not been defined before.

Our motivation to refer to our Eq. 4 as the Carnot limit of a "cold" heat engine is similar to a cold car engine in winter. When the car engine is still cold in winter just after it has been started, one needs to hit the gas harder to get the same power. Our expression captures this effect: The heat flux needs to be larger to get a certain power, because the term $dU_a/dt$ reduces the effect of the heat flux on the power. As this is a storage effect similar to a cold car engine heating up, we think that the term "cold heat engine" nicely captures this storage effect.

In the revision, we will motivate and clarify the justification for the "cold" heat engine terminology in greater detail in the introduction and section 2.1.

*Reviewer comment 2: From Fig. 1 it seems that the heat engine discussed by the authors is confined to the radiative-convective layer with $J_{out}$ being a flux into the free atmosphere above. However, combining Eqs 1,4,5 gives $J_{out} = R_{l,out} - R_{l,net}$, i.e. the cooling of the whole atmospheric column by thermal radiation. Thus, either it is assumed that there is no exchange between the radiative-convective layer and the free atmosphere, or the heat engine comprises the whole column. This needs to be clarified (in Fig.1 and/or the text introducing the heat engine).*

**Reply:** We apologise, as it appears that Fig. 1 is misleading in this respect. $J_{out}$ does not actually go into the free atmosphere, but rather stays in the radiative-convective layer and is exported to above by net emission of longwave radiation. In other words, the radiative-convective layer is heated by the turbulent heat fluxes from the surface ($J_{in}$), and the cooling takes place not by a heat flux, but net longwave radiative cooling of the radiative-convective layer, and this cooling is represented by $J_{out} = R_{l,out} - R_{l,net}$, as you write.

We will clarify this aspect in the revision.

*Reviewer comment 3: Eq. 2 gives the entropy budget of the heat engine. However, $J_{out} = R_{l,out} - R_{l,net}$ (see above), and $R_{l,net}$ is the sum of thermal flux coming from the atmosphere (approx. $R_{l,out}$, say) and from the surface ($R_{l,surf}$). Thus, instead of*

$J_{out}/T_a$ I would expect a term ($R_{l,surf}/T_s$) and something like $2R_{l,out}/T_a$ appearing in Eq. 2, representing both the import of entropy from the soil and the respective export from the atmosphere. It seems that $R_{l,surf}/T_s$ can be of the same order as $J_{in}/T_s$. The authors need to explain why the entropy import from the surface ($R_{l,surf}/T_s$) is not considered, in particular as $J_{out}/T_a$ is used to obtain Eqs. 3,4,7.

**Reply:** Thank you for pointing this out. The important point that we did not describe well enough in the text is that the entropy budget expressed in Eq. (2) is the budget for thermal entropy, not for radiative entropy. This is an important distinction. What you describe as terms such as $R_{l,surf}/T_s$ represent terms of radiative entropy, i.e., it is entropy reflected in the composition of radiation, but not associated with heat (or thermal energy). As we deal here with convection and a heat engine, we must not include radiative terms as such, but only when radiation is absorbed and heats (adds thermal energy), or when the net emission of radiation cools (removes thermal energy). Radiative terms and radiative entropy production are typically much larger in the Earth system than non-radiative contributions (easily by a factor of 100). Yet, any form of motion is associated with the much smaller, but relevant thermal entropy terms.

In the revision, we will explain this important point.

***Reviewer comment 4:*** *Eq. 7 gives an estimate for $J_{in}$ derived from optimization based on the second law. However, using Eqs. 1,5 to replace $dU_a/dt$ and $dU_s/dt$ in Eq.7 (or replacing $dU_s/dt$ in Eq. 5 by $J_{opt}$ with $dU_a/dt$ as described in Sec. 2.4) shows (if I'm not wrong) that $J_{opt}$ is not equal $J_{in}$. Thus, while $J_{opt}$ results from utilizing the second law it seems not to be consistent with energy conservation (the first law) within the same model framework (Eqs 1,5). If the conclusion (and the approach taken) that the turbulent fluxes optimize the heat engine constrained by energy conservation holds, this surprises me. What is the explanation (perhaps it is trivial)?*

**Reply:** Thank you for pointing out this discrepancy. It actually points out an error in Eq. 1 when dealing with the atmospheric energy budget. It has to do with the term

$R_{l,net}$, i.e., the net radiative cooling of the surface. In the optimisation, the outcome of $J_{in} = J_{opt}$ results in $R_{l,net} = R_{s,avg}/2$. This net emission of radiation at the surface may be absorbed within the radiative-convective layer, or it may pass this layer to higher levels, depending on the optical thickness/absorptivity of the layer. If it is re-absorbed, then it adds a radiative heating term to Eq. 1, and it adds a radiative heating term to the entropy budget (Eq. 2). However, as this radiation is absorbed at the prevailing physical temperature in the atmosphere, rather than the potential temperature associated with adiabatic motion, it is likely to be absorbed closer to $T_a$ than to $T_s$. So it adds a term $R_{l,net}/T_a$ to the entropy budget. When combining the entropy budget with Eq. 1 to replace $J_{out}$, the terms involving $R_{l,net}$ drop out, resulting in no change in Eq. 3 and 4. If it passes the layer without being re-absorbed, then it does not affect Eqns. 1 - 4.

We therefore suggest to change the formulation in section 2.1 slightly in the revision and refer to $dU_e/dt$ instead of $dU_a/dt$ in Eq. 1, i.e., the heat storage change inside the engine, rather than the lower atmosphere. The reason for this renaming is that section 2.1 deals with the derivation of the limit in the presence of heat storage changes, and it is only afterwards that radiation and other aspects are introduced. Also, the term $R_{l,net} = R_{s,avg}/2$ in the case of maximum power is actually quite small. We would add this discussion to an Appendix.

***Reviewer comment 5:*** *The difference between $J_{opt}$ and $J_{in}$ (as explained above) is given by $R_{l,net} - R_{l,out}/2$ (again, I hope that I'm not wrong). In Fig 3, although it is hard to judge, this difference seems to be relatively large, and larger than the difference between $J_{opt}$ and $J_{obs}$. If so, this surprises me too. Perhaps, the authors may like to compute this $J_{in}$ (consistent with energy conservation constraint), compare it with $J_{opt}$ ($J_{obs}$), and discuss the result in the context of the optimization concept.*

**Reply:** Actually, as explained in the previous comment, $R_{l,net} = R_{s,avg}/2$ in the optimised case, so that the expression $R_{l,net} - R_{l,out}/2$ is actually zero, so that there is actually no discrepancy between $J_{opt}$ and $J_{in}$. So we do not think that we need to explain something here (except, perhaps, that we do not reproduce the slight diurnal

variation in $R_{l,net}$ with our approach).

***Reviewer comment 6:*** *I do not understand Fig 2a: A more comprehensive explanation may be given in the text: e.g. what defines the particular shape of the atmospheric heat storage change (pink area).*

**Reply:** We apologise for not explaining this in more detail. What is shown by the pink area is the typical change of the temperature profile in the lower atmosphere. It warms during the day with an adiabatic lapse rate (i.e., the linear decrease with height shown by the dashed line), while at night, the lower atmosphere cools, and often it cools more near the ground, resulting in a night-time inversion (shown by the other dashed temperature profile).

In the revision, we will provide this explanation in the text in comprehensive form.

***Reviewer comment 7:*** $R_s -> R_{s,ave}$.

**Reply:** Thanks, we will adjust this in the revision.

---

## Editor Comment (EC1) · M. Crucifix (Editor) · 20 Jul 2018

Thank you for your thoughtful review of the manuscript. Based on your comments, the comments of the other reviewer, and the responses of the authors, the latter will be suggested to upload the revised version of their manuscript for final evaluation by myself.

---

## Author Response (AR1)

**Author Response to Manuscript "Diurnal land surface energy balance partitioning estimated from the thermodynamic limit of a cold heat engine" by Kleidon and Renner**

Dear Editor:

We would like to submit our revised manuscript to Earth System Dynamics. In the revision, we addressed the main points raised by the reviewers along the lines described in our response in the discussion forum (apart from a few, minor cosmetic changes). In the submitted manuscript, we highlighted the modified text passages in blue, so that they are easy to identify.

The points raised by the reviewers were dealt with as follows:

1. **Better motivation/explanation for the term "cold heat engine" (response to Reviewer 1 Comment 1 and Reviewer 2 Comment 1)**

   We explain our motivation for referring to the limit derived in the manuscript as the limit of a cold heat engine in greater detail and made it clearer that this is a new term, and not an established term. We changed part of the abstract to clarify this, and added text at the end of page 2 in the introduction (using the explanation from the discussion forum) and at the end of Section 2.1 on page 6.

   We also included Reviewer 1's point that despite the difference between where the heat storage change is located, the power output of the heat engine is the same (page 7, line 24).

2. **Confirm derived power with data (Reviewer 1 Comment 2)**

   We included this point as a possibility for future research in the discussion section (page 11, line 24-26).

3. **Emphasize utility of Eq. 2 (Reviewer 1 Comment 3)**

   We followed Reviewer 1's suggestion and emphasised more the utility of the simple estimate for turbulent heat fluxes. To do so, we first numbered the two approximations (Eq. 8 and 9 on page 7) and describe the constant offset, as described by the reviewer. In the discussion section, we also refer to the utility of this estimate (page 11, line 16-17).

4. **Restriction of applicability of the approach for values Rs < Rs,avg/2 (Reviewer 1 Comment 4)**

   During the revision, we actually noticed that the argumentation of the reviewer was not quite correct. The expression for power does not turn negative for Rs < Rs,avg/2, but the power always stays fixed at a value of Rs,avg/2. It is thus not a physically implausible case, and so we decided to not change the text in the manuscript. (That the approach likely does not apply during nighttime conditions due to stability (and lack of heating) is already discussed in the discussion section).

5. **Fix text (Reviewer 1 Comment 5)** — Done

6. **Explain blue boxes in Figure 2 (Reviewer 1 Comment 6)** — Done

7. **Enlarge Figure 3 for visibility (Reviewer 1 Comment 7)** — Done

8. **Setup of heat engine shown in Figure 1 (Reviewer 2 Comment 2)**
   We modified Figure 1 so that the heat flux Jout does not go into the free atmosphere, but rather into the radiative-convective layer.

9. **Improved description of entropy budget (Reviewer 2 Comment 3)**
   We extended the description of the entropy budget (Eq. 2) in Section 2.1 (page 5, lines 12 - 19) to address this point.

10. **Difference between Jin in Eq. 1 and Jopt (Reviewer 2 Comment 4)**
    As described in our reply in the discussion forum, this apparent difference is due to a somewhat altered atmospheric energy budget (and thus an altered formulation of the first law) that needs to account for Rl,net. We renamed the heat storage change inside the engine to $dU_e/dt$ in Sect. 2.1 and added an Appendix in which we describe that the limit derived in Sect. 2.1 still applies when the net exchange of longwave radiation between the surface and the atmosphere is added in the derivation. We also included $dU_e/dt$ in Table 1.

11. **Calculate difference between Jopt and Jin (Reviewer 2 Comment 5)**
    As described in our response, there is no difference between these two expressions (related to the term Rl,net, related to Comment 4 of Reviewer 2). We hence did not change the manuscript.

12. **Improved explanation of Figure 2a regarding the shape of storage changes (Reviewer 2 Comment 6)**
    We added a reference to Figure 2A in Sect. 3 and added a brief explanation of the profiles in the caption of Figure 2.

With these modifications, we hope that we addressed all points raised by the reviewers in a satisfactory way.

Best regards,

Axel Kleidon
on behalf of the authors